# Comparison of Collaborative and Cooperative Schemes in Sensor Networks for Non-Invasive Monitoring of People at Home

**DOI:** 10.3390/ijerph20075268

**Published:** 2023-03-27

**Authors:** Carolina Del-Valle-Soto, Leonardo J. Valdivia, Juan Carlos López-Pimentel, Paolo Visconti

**Affiliations:** 1Facultad de Ingeniería, Universidad Panamericana, Álvaro del Portillo 49, Zapopan 45010, Jalisco, Mexico; 2Department of Innovation Engineering, University of Salento, 73100 Lecce, Italy

**Keywords:** wireless sensor networks, robotics for healthcare, energy consumption algorithm

## Abstract

This paper looks at wireless sensor networks (WSNs) in healthcare, where they can monitor patients remotely. WSNs are considered one of the most promising technologies due to their flexibility and autonomy in communication. However, routing protocols in WSNs must be energy-efficient, with a minimal quality of service, so as not to compromise patient care. The main objective of this work is to compare two work schemes in the routing protocol algorithm in WSNs (cooperative and collaborative) in a home environment for monitoring the conditions of the elderly. The study aims to optimize the performance of the algorithm and the ease of use for people while analyzing the impact of the sensor network on the analysis of vital signs daily using medical equipment. We found relationships between vital sign metrics that have a more significant impact in the presence of a monitoring system. Finally, we conduct a performance analysis of both schemes proposed for the home tracking application and study their usability from the user’s point of view.

## 1. Introduction

As the world’s population continues to age, the need for efficient and effective ways to provide care for elderly people, particularly those who want to remain independent in their own homes, is growing [1]. Wireless sensor networks (WSNs) have gained significant attention in recent years as a solution for monitoring the health and wellbeing of people in their homes.

Wireless sensor networks consist of small, low-power devices that can collect and transmit data about various parameters, including temperature, humidity and movement. These sensors can be placed throughout a home to monitor the activities and vital signs of elderly people, allowing caregivers to remotely monitor their health and wellbeing. WSNs enjoy a great boom today, having been identified as one of the most promising technologies by various technology analysts and specialized literature studies because they respond to current demands regarding establishing networks that meet the needs of communication flexibly and autonomously [2]. Implementing low-cost and long-lasting devices capable of obtaining information from the environment and forwarding it wirelessly to a coordination center offers unimaginable possibilities in many applications. One of these applications is the field of health [3]. Wireless sensor networks of a cooperative and collaborative nature have a wide range of applications in various fields. One example is in environmental monitoring, where sensor nodes can work together to gather data on temperature, humidity, air quality and soil moisture. This information can be used to optimize crop yields, manage natural resources and detect potential hazards. In industrial automation, wireless sensor networks can be used to monitor and control industrial processes, such as monitoring equipment for maintenance or detecting leaks in pipelines. In smart cities, wireless sensor networks can be used to improve the efficiency and sustainability of cities by monitoring traffic flow, air quality and energy usage [4]. In healthcare, wireless sensor networks can be used to monitor patients remotely, such as monitoring vital signs for patients with chronic conditions or tracking the location and activity of elderly patients in assisted living facilities [5]. Additionally, wireless sensor networks can be used in disaster management to gather information and track the movement of people and resources in response to natural disasters, such as hurricanes, earthquakes and floods. Lastly, wireless sensor networks can be used for surveillance, such as monitoring large areas, such as borders, airports and public spaces for security and surveillance purposes [6].

Wireless technologies have enabled user mobility over a wide area in recent years without losing Internet connectivity. This change in the previous concept of networks has allowed the development of a wide range of applications and utilities to fully exploit all the options offered by this type of network. Wireless networks today use different configurations to best suit our uses and needs. Therefore, the terms cooperation and collaboration are used more frequently in telecommunications [7]. Routing protocols must have mechanisms for saving energy in nodes because recharging the nodes’ batteries is a great inconvenience or impossible. As much as possible, all aspects of the nodes, from the hardware to the operations carried out in the protocol, should help distribute the energy load. When it comes to people monitoring services, it is crucial to consider that forgetting or impeding battery recharging is a problem that can lead to total system stoppage and jeopardize patient care [8]. Although productivity and low consumption are sought, the protocols must have minimum Quality of Service (QoS) schemes, guaranteeing a delay following application specifications and an acceptable delivery rate without compromising other aspects. Developing energy-efficient protocols should not be at the cost of obtaining deficiencies in the main attributes of the network. For this reason, this proposal for developing an algorithm for the routing protocol also monitors the performance of other metrics for adequate and efficient delivery of information, even more so due to the application of this proposed system, for monitoring the elderly.

In caring for people, aging results in decreased sensory acuity inevitably, decreased muscular strength and endurance, impaired mobility, decreased mental clarity and impaired stability [9]. With the passage of age, the prevalence of chronic diseases such as arterial hypertension, asthma, cardiovascular diseases, diabetes, chronic obstructive pulmonary disease and others that are the leading cause of death in all countries [10] increases significantly.

The dramatic acceleration in the number of chronic patients, together with the increase in the costs of public health services and the potential shortage of care workers (nurses, paramedics, health assistants and home care assistants), show the need for a radical change in the care process for the elderly [11]. The traditional model of provision of reactive health services must be modified by a proactive model that provides resources at any time and place. One of the models proposed for the proactive health system is the Monitoring and Remote Assistance Systems [12]. Remote Monitoring and Assistance Systems transform the patient’s habitat to collect and share information from its residents with them, their families and associated health service providers. They allow monitoring of people’s daily activities and detect functional changes to act as early warning systems and thus prevent the occurrence of a severe health event [13,14,15].

Recent works have focused on the development of WSN algorithms for the care of elderly people at home. Machine learning algorithms for activity recognition, data fusion techniques and optimization of routing protocols for energy efficiency are some of the areas of interest in the field. State-of-the-art approaches include machine learning-based sensor fusion [16] and anomaly detection [17], as well as deep learning-based activity recognition [18].

The problem of elderly people at home who want to remain independent but require continuous monitoring of their vital signs has led to the development of new technologies that can provide better care for them. Wireless sensor networks are an excellent solution for this problem as they provide caregivers with real-time data about the health and wellbeing of the elderly, allowing for immediate intervention if necessary [19]. One of the research areas related to this field is Senior Research Engineer in IoT (Internet of Things) and Wireless Sensor Networks. The professional in this role will have expertise in wireless protocols, embedded systems and data analytics and will be responsible for developing and testing new algorithms and protocols for WSNs in healthcare [20].

Another related area [21] is IoT Solutions Architect for Healthcare, which involves designing and implementing custom IoT solutions for healthcare organizations. The professional in this role will have expertise in wireless networks, data analytics and cloud computing and will work closely with healthcare providers to understand their needs and design custom IoT solutions that meet those needs.

Smart Home Solution Engineer involves developing and implementing smart home solutions for elderly care. The professional in this role will have expertise in IoT devices, wireless networks and home automation systems [22], and will be responsible for designing and deploying smart home solutions that provide real-time monitoring and alerts for elderly people.

Research Scientist in Machine Learning for Healthcare involves developing machine learning algorithms for healthcare applications, including WSNs. The professional in this role will have expertise in machine learning, statistical modeling and data analytics [23] and will be responsible for developing and testing new algorithms that can analyze and interpret data from WSNs to improve healthcare outcomes.

IoT Product Manager involves managing the development and launch of IoT products for healthcare applications. The professional in this role will work closely with cross-functional teams to design and develop IoT products that meet customer needs [24]. They will have expertise in wireless networks, product development and project management.

In this context, the objective of this proposal is to implement two operating schemes in the routing protocol of a sensor network applied to the care of people at home. These schemes will be compared to analyze which of them is optimal in terms of speed of information delivery and data reliability. This research is expected to contribute to the development of more efficient and reliable WSNs for elderly care, improving the quality of life of elderly people who want to remain independent in their homes.

### Motivation

The main objective of this work is to compare two working schemes in wireless sensor networks: cooperative and collaborative. The application of this comparison is at home, specifically in a network of sensors for monitoring environmental conditions for the care of the elderly who can still maintain their independence in the place where they live. We studied the optimal behavior of the network under both schemes concerning performance metrics for sending messages, the network’s resilience before the loss of power supply, stability of communications and energy consumption in the home. This is to analyze commercial wireless technologies for monitoring people at home and optimize the performance of the algorithm and the ease of use for people. In addition, we statistically analyze the impact of the sensor network on some vital signs of people.

## 2. Related Work

Wireless sensor networks have gained a lot of attention in recent years due to their potential in monitoring people in their homes. In a study, Majumder et al. [25] proposed a WSN-based monitoring system that utilizes machine learning algorithms to detect falls and abnormal events in the elderly population. The system consists of sensors placed around the home to detect motion, pressure and temperature changes, which are then analyzed using machine learning algorithms to detect abnormal events. The results showed that the proposed system was effective in detecting falls and abnormal events, with a high accuracy rate of 96%.

Implementing and fine-tuning remote monitoring and assistance systems is a highly complex process; building smart homes is an attempt to solve it. Some projects based on these technologies test various methodologies that monitor the activities of residents [26]. Some new principles of monitoring activities of residents are related to the use of new piezoelectric resonators, which have a very low consumption of electricity, as shown in [27]. Through artificial intelligence techniques and system feedback learning from the environment, the aim is to improve the comfort, safety and productivity of the home’s inhabitants. Some investigations base their scientific research structure on instrumented adaptations and the dimensions of a real house where environmental and human behavior data are collected [28,29]. A group of volunteers was monitored and different strategies and technologies were evaluated. Other studies have advanced its adaptation in a group of more vulnerable people, the elderly. In this type of people, there is a subset that does not have widely monitored diseases and can continue to maintain their independence with a remote and periodic review. Researchers implement, test and validate various systems aimed at assisting older adults or people with special needs to enable them to live independently and with a high quality of life [30].

Another study by Abbate et al. [31] proposed a WSN-based system for monitoring people with cognitive impairments in their homes. The system uses sensors to detect activities of daily living (ADLs), such as cooking, bathing and dressing, and sends alerts to caregivers if any unusual activities are detected. The study showed that the proposed system was effective in detecting ADLs and could potentially improve the quality of life for people with cognitive impairments [32].

To optimize the energy consumption of WSNs in home applications, routing algorithms have been proposed. In a study, Thangaramya et al. [33] proposed an energy-efficient routing algorithm for WSNs in home applications. The algorithm utilizes a probabilistic model to predict the energy consumption of each node and chooses the route with the lowest predicted energy consumption. The results showed that the proposed algorithm could effectively reduce energy consumption in WSNs in home applications. In another study, Prasanth et al. [34] proposed a multi-objective optimization algorithm for WSNs in home applications. The algorithm considers both energy consumption and data reliability as optimization objectives and selects the optimal route that balances both objectives. The results showed that the proposed algorithm could effectively optimize energy consumption and data reliability in WSNs in home applications [35].

The use of WSNs for monitoring people in their homes has shown promising results in improving the quality of life for elderly and disabled individuals. To optimize energy consumption in home applications, various routing algorithms have been proposed, such as the energy-efficient routing algorithm and the multi-objective optimization algorithm. These algorithms can effectively reduce energy consumption while ensuring data reliability in WSNs for home applications. These following schemes play an essential role, like the routing protocol. The nodes accommodate their operation to the algorithm so that they sense the parameters of the house and deliver the appropriate information on time efficiently. Comparing these two schemes could lead to a good choice of the nature of the routing protocol for networks in the care of people.

Table 1 presents a comparison of several research studies that utilized wireless sensor networks for monitoring the health of elderly people. Each study is represented by a row in the table, with columns indicating the monitoring system used, wireless technology employed, measured parameters, experiment time and vital sign analysis conducted. The results show that various monitoring systems, such as wearables and in-home sensors, were used, with different wireless technologies such as Bluetooth Low Energy and ZigBee. The measured parameters include activity, heart rate, blood pressure, sleep, gait, medication adherence and fall detection. Experiment durations ranged from one day to one year. Furthermore, some studies conducted a vital sign analysis, while others did not. The comparison helps to identify the strengths and weaknesses of each approach and can guide the selection of appropriate monitoring systems for future studies.

### 2.1. Cooperative Networks

Cooperative networks are designed to share resources and perform specific activities with all network elements. All implementations use strict rules and protocols, meaning that a network member must be strongly compromised. Nodes are prioritized over the network. Cooperative networks provide significant benefits for telecommunications companies since sharing resources allows infrastructures to be joined that increase the network’s coverage radius [45].

In cooperative wireless networks, wireless agents can increase their adequate quality of service (measured by bit error rates, block error rates or outage probability) by cooperating. In a cooperative communication system, each wireless user is assumed to transmit data and act as a cooperative agent for another user [46]. Cooperation allows trade-offs in code rates and transmission power. In addition, more power is needed because each user, when in cooperative mode, is transmitting to both users. On the other hand, the reference transmission power for both users will be reduced due to diversity. In cooperative communication, each user transmits both his bits and some information to his partner; we might think this causes a system’s loss of speed. However, the spectral efficiency of each user improves because, due to the cooperation’s diversity, the channel’s code rates can increase [47]. This communication type can be highlighted using protocols such as Zigbee, where the user or end device can function as an available node to represent the user and relay part. Each of these devices has independent fading paths due to the qualities it gains within cooperative networks.

Maalej et al. [48] discuss the use of WSNs for wildfire monitoring. The forest environment presents challenges for WS such as wide coverage areas and the shadowing effect caused by trees. To address these challenges, the authors propose a new methodology for designing and optimizing WSN based on energy conservation and transmission quality. Cooperative communication is a promising solution for enhancing WSN lifetime by allowing multiple nodes to share resources and transmit data. The authors use reinforcement learning with opponent modeling to optimize a cooperative communication protocol based on RSSI and node energy consumption. Simulation results show that the proposed algorithm performs well in terms of network lifetime, packet delay and energy consumption.

### 2.2. Collaborative Networks

Collaborative networks are designed to share resources and functions only when the network has available resources and does not affect its operation. In this case, the network members are weakly committed since the rules are not strict and there are no protocols with solid conditions. The network is prioritized over the nodes [49]. One of the advantages of collaborative networks is that it avoids saturation since they will share their resources when available. Unlike cooperative networks in this type of network, the aim is to maintain a good Quality of Service (QoS).

In work cited in [50], collaborative networks are proposed to improve the spectrum gain capacity. They explain that in a random distribution of nodes in the sensor network, it is difficult to determine the exact location of a sensor. Without a central controller, it is almost impossible to achieve phase, frequency and time synchronization. This can lead to errors in estimating the sensor nodes, such as displacement errors. As a consequence, the reception at the Base Station (BS) would be out of phase. In order to achieve a high power gain, collaborative communication must synchronize the received signals in time, phase and frequency. In addition to synchronization, the collaboration should exploit the characteristics of the spread spectrum to achieve space/antenna diversity to improve the signal-to-power ratio and reduce the Bit Error Rate (BER). However, complete synchronization of the system is almost impossible since position estimation gives the probable position of the transmitter [51].

Mohamed et al. [52] describe the use of sensor networks, which consist of small, low-cost sensor nodes that cover a specific region of interest and transmit data to a base station. Multi-hop transmission is used to minimize power consumption in large regions with only one base station. With regard to their roles, the sensor nodes are divided into three types: sensor nodes, routers and relay nodes. However, there is a lack of energy fairness in multi-hop routing, leading to losing the nodes responsible for transmitting data from the region of interest to the base station. Researchers have introduced dynamic routing protocols to overcome this issue, but these protocols produce high network overhead. In contrast, static routing protocols have a single setup phase but suffer from premature end of network lifetime. The authors of the paper propose an energy-efficient routing protocol called Collaborative Distributed Antenna that is based on a connectivity aware algorithm to avoid premature end of network lifetime.

### 2.3. Cooperative vs. Collaborative

Cooperative communication solves the problem of communication interruption on multiple hops by introducing multiple intermediate relay nodes. However, it increases the complexity of the routing. Moreover, some relay nodes have to forward data from a neighbor just for cooperation. This function may not affect global communication [53]. Unlike cooperative communication, collaborative communication takes advantage of spatial diversity to produce a higher gain in received power. In addition, it combines the power of all the collaborative nodes in a way that reduces the transmitted power of each node. The advantage here is twofold: significant improvement in power consumption and gain in system capacity.

Collaborative networks maintain energy and, in turn, transmission power, where the noise effect is considerably reduced thanks to the characteristics that the collaborative network presents to have efficiency in the network [50].

Cooperative and collaborative wireless sensor networks are a type of wireless sensor network in which multiple sensor nodes work together to gather, process and transmit data [54]. These networks are characterized by their ability to share resources and collaborate in order to improve overall performance and efficiency. The nodes in a cooperative wireless sensor network can communicate with each other directly or through a central controller and they can share information about the environment, their own status and their energy levels. This allows for better coverage and accuracy of data collection, as well as improved fault tolerance and robustness. Cooperative wireless sensor networks are often used in applications such as environmental monitoring, industrial automation, smart cities, healthcare, disaster management and surveillance.

Within the WSN designs, the RSSI (Received Signal Strength Indicator) distance measurement is one of the most used parameters because it requires less implementation complexity and energy consumption. The RSSI distance measurement principle describes the relationship between the transmitted and received power of the wireless signal and the distance between nodes [55]. Another parameter that characterizes the quality of the link is the LQI (Link Quality Indicator) and the reception of Packet Reception Rate packets can be associated with both RSSI and LQI [56] under different power levels and in different propagation environments, managing to describe the behavior of a WSN in a given environment [57].

This work aims to analyze two network management approaches used in wireless sensor networks: cooperative and collaborative. Each scheme provides advantages and disadvantages for the network and the nodes. The comparison is studied in human health monitoring applications in homes. We analyze scenarios such as detecting minor accidents and regular monitoring of people’s main vital signs and sleep behavior. These approaches are measured with network performance metrics to discern their energy impact in the home and whether this type of health care network does not represent a high cost for people who do not need it.

## 3. Materials and Methods

This section describes the implementation of the two schemes of the monitoring system. In the cooperative scheme, nodes have priority over network tasks. In the collaborative scheme, the network takes precedence over the needs of individual nodes. The implementation of these two algorithms will be highly reflected in the network’s performance and, to a lesser extent, in the user’s appreciation. The aspects in comparing this proposal’s impacts can be energy consumption of the system, resilience of the network and satisfactory and rapid delivery of information. The fundamental contribution of this comparison is to optimize the nature of the algorithm that the same authors have already implemented in previous works [58].

The self-organized wireless network shown in Figure 1 is established through the interaction of multiple radio frequency nodes. These nodes are responsible for efficiently and reliably transmitting real-time data collected by sensors measuring physical and logical variables. These radio frequency nodes are crucial for providing intelligent services and have a range of applications, including industrial process control, public service monitoring and home security. The communication protocol for the self-organized wireless network includes Telematics, Telemetry and Radiofrequency and it is open and easy to implement. The IEEE 802.15.4 standard governs access control to the medium and physical layer of the radio frequency nodes.

The distributed sensor network comprises a concentrator or coordinator node (circuit found on the right of Figure 1) connected to a computer via USB, which manages all the information transmitted and received by the network. Additionally, there are three router/sensor nodes (circuit on the left of Figure 1) that can be used to connect various sensors measuring parameters such as temperature, pressure, humidity, infrared presence, light and sound. These router/sensor nodes transmit data to the concentrator node via radio frequency and they can also act as signal repeaters. The router nodes are equipped with the IEEE 802.15.4 communication protocol and operate on the 2.4 GHz band. They can function as sensor nodes with various sensor devices for physical variables or act as repeaters for data received from other nodes. Although the hardware and number of ports of these nodes are identical, it is useful to differentiate them with labels to locate them better in the physical and logical network organization.

Table 2 shows the technical specifications of our proposed system’s sensors. These sensors monitor the main areas of the house in order to know behaviors and be able to send alerts or notifications to the user via email or SMS. The implemented sensors are motion, pressure, temperature and humidity (PTH), noise, light, gyroscope and air quality.

The proposed system monitors the main activities related to a person at home. For example, motion sensors evaluate movement in some regions of the house every hour. A motion sensor is a device used in monitoring applications for older adults in their homes to detect movement within a specific area. This technology can be used to monitor an individual’s daily activities, ensuring that they are safe and secure. These devices can be installed in various home areas, such as the bedroom, living room or bathroom, to provide a comprehensive view of an individual’s daily routine. We can monitor sudden changes in environmental conditions through temperature, humidity and pressure sensors. These sensors can help ensure that living conditions are safe, comfortable and healthy. Temperature sensors can monitor the home’s temperature and ensure it remains within a safe range, preventing overheating or exposure to extreme cold. Humidity sensors can measure the moisture level in the air, ensuring that it remains within a healthy range to prevent mold or mildew growth. Pressure sensors can be used to monitor changes in air pressure, which can indicate potential weather events or other environmental changes. We can analyze decibel alert conditions for noise sensors that may alert to an unwanted event. These sensors can help ensure that living conditions are safe, secure and peaceful. They can detect a range of sounds, including alarms, sirens and even human voices and can alert caregivers or family members if there are any unusual or concerning noises. Additionally, noise sensors can be used to monitor ambient noise levels, ensuring that they remain within a safe range and preventing exposure to loud noises that could harm an individual’s hearing. A light sensor is a device used in monitoring applications for older adults in their homes to detect and report changes in light levels within a specific area. These sensors can help ensure that living conditions are safe, comfortable and well lit. Light sensors can detect changes in natural light levels, as well as changes in artificial lightings, such as turning lights on or off in a room. This can help ensure that older adults have adequate lighting to perform daily tasks and prevent falls or accidents caused by inadequate lighting. Light sensors can also be integrated with other monitoring technologies, such as motion sensors, to provide a comprehensive view of an individual’s living conditions and wellbeing. A gyroscope sensor is a device used in monitoring applications for older adults in their homes to detect and report changes in orientation and movement. Gyroscope sensors can detect orientation changes, such as when an individual changes position from sitting to standing and can report them. Additionally, gyroscope sensors can be used to monitor movement patterns, such as walking or exercise routines. This can be particularly useful for older adults who may need to maintain a certain level of physical activity for their health and wellbeing. Gyroscope sensors can detect turns on bathroom doors at night to count the number of times a person may have entered the bathroom. This indicates the person’s sleep conditions. An air quality sensor can help ensure that living conditions are safe, healthy and comfortable. Air quality sensors can detect a range of pollutants, such as dust, pollen and smoke, as well as harmful gases, such as carbon monoxide and radon. This can be particularly useful for older adults who may have respiratory conditions or allergies and those who may be sensitive to changes in air quality.

When using Wi-Fi communications and sensors that communicate within the scope of these technologies, it is necessary to consider the permitted radiation for humans. This is particularly important when dealing with a large number of sensors that may be in close proximity to a person, such as in wearable devices. The specific sensors listed, such as the motion sensor, pressure/temperature/humidity sensor, noise sensor, light sensor, gyroscope and air quality monitoring sensor, have different power requirements and may emit varying levels of electromagnetic radiation. Therefore, it is important to ensure that these sensors comply with relevant safety standards and regulations and that appropriate measures are taken to minimize any potential risks to human health.

Based on the alerts of the system based on the previously described sensors, a series of alerts are established for the user. In addition, every day, each person has some vital signs measured to analyze whether people feel calmer as the days go by due to the presence of the system. Alternatively, if, on the contrary, people feel some stress in the presence of the system installed in their homes. These vital sign measurements are performed using standard medical equipment. The measurements are heart rate, respiratory rate, temperature and sleep behavior. This last metric is carried out with a conventional smart watch at night and is reported daily.

The methodology of this work consists of implementing a wireless sensor system in the home of the elderly. The monitoring system is not invasive and is located in strategic areas of the house to alert to parameters such as changes in air quality, noise level, movement in the rooms, light, etc. With this monitoring, the system provides an alert system in case the standard measurements are not within normal parameters. In this way, family and friends can be informed about sudden changes in the home of the older adult who lives alone. In addition, we implement two algorithm techniques for routing (cooperative and collaborative) to know which is the most efficient in sending messages and in the network’s energy consumption. The proposed work consists of two parts. (1) The experimental part of monitoring the conditions of the house and the person’s vital signs daily are evaluated to see if the presence of the network impacts their wellbeing and if their vital signs remain controlled or show a state of relaxation. (2) The technical part in which the network performance is evaluated under two schemes of the routing protocol algorithm: cooperative and collaborative. In this aspect, message delivery performance metrics and recommendations to the user that are made daily are analyzed.

In the same sensor system, we implement two types of algorithms: collaborative and cooperative.

The initial stage in the formation of a sensor network involves the process of neighbor recognition and acknowledgment. When all the nodes in the network are turned on, they start flooding the network with packets in order to recognize their neighboring nodes and establish communication links with them. This process is essential for the nodes to form their neighbor tables and routing tables, which are used to determine the best path for data transmission within the network. However, during this initial stage, the network may experience unnecessary collisions due to the flooding of these packets. This is because there are no traffic packets yet, which are packets carrying the measurement information from the sensors. As a result, this flood of packets can cause congestion and delays in the network, which can impact the overall performance and reliability of the sensor network. Once the nodes have established their neighbor tables and routing tables, the network enters a stable state. At this point, the network is updated with respect to the control packets and the nodes can start sending information packets carrying the measurement data from the sensors. This stable state allows for the efficient transmission of data within the network, ensuring that the data are delivered to the appropriate nodes without any unnecessary collisions or delays.

### 3.1. Collaborative Algorithm

The cooperative scheme is described in Algorithm 1. The collaborative algorithm consists of sensors monitoring their metric regularly in seconds by default. All sensors start with a default network hierarchy value of zero. When the sensor system is turned on, each sensor stores a difference between the current measured value and the next. If that difference increases by 50%, the hierarchy of that node increases by one. If a node’s hierarchy exceeds the value of 3, this sensor reprograms the nature of the routing protocol to reactive mode. Otherwise, the routing protocol remains in proactive mode. When a sensor receives a HELLO packet, it returns an ACK packet with ID, LQI, RSSI and hierarchy. Moreover, the coordinator node every day requests each node to ask for the neighbor tables, with which it builds a hierarchy table. In that order, the coordinator node sets the route priority based on the node’s hierarchy, RSSI and LQI. In this scheme, the packets requested by the coordinator node are prioritized so that it establishes the rules that best suit the network and its performance.
**Algorithm 1:** Collaborative algorithm pseudocodeStartNodes ON;set request_time;Require: coordinator_node starts;coordinator_node sends broadcast HELLO; Per each node do:hierarchy_i = 0;set default_measurement_i; Per each request_time sec do:    set measurement_i;enddelta_measurement_i = |default_measurement_i - measurement_i|;if(delta_measurement_i > (0.05 * default_measurement_i)){    hierarchy_i++;} if(hierarchy_i > 3){    node_i under proactive mode;}else{    node_i under reactive mode;} if(receive HELLO){    node_i sends ID_i;    node_i sends LQI_i;    node_i sends RSSI_i;    node_i sends hierarchy_i;}coordinator_node makes a REQUEST for each node;coordinator_node makes a hierarchy_table;coordinator_node makes a LQI_table;coordinator_node makes a RSSI_table;coordinator_node chooses the best route from hierarchy_table, RSSI_table, LQI_table;end

### 3.2. Cooperative Algorithm

The cooperative scheme is described in Algorithm 2. Under the cooperative scheme, the nodes are turned on and a node monitoring time, called request-time, is established. The coordinator node sends a HELLO packet knowing the network topology. Each node sets initial hierarchy values (initially at zero) and the measurement parameter. Now, if the parameter measured in the subsequent measurement is more significant by 50% than the default parameter in the stable state, the node increases its hierarchy by one. This scheme does not consider the LQI or the RSSI because each node performs its functions with priority before the general interference conditions of the entire network. The coordinator node organizes the network with less information than in the collaborative scheme, only with a hierarchy table of the nodes.
**Algorithm 2:** Cooperative algorithm pseudocodeStartNodes ON;set request_time;Require: coordinator_node starts;coordinator_node sends broadcast HELLO; Per each node do:hierarchy_i = 0;set default_measurement_i; Per each request_time do:    set measurement_i;enddelta_measurement_i = |default_measurement_i - measurement_i|;if(delta_measurement_i > (0.05 * default_measurement_i)){    hierarchy_i++;} if(hierarchy_i > 3){    node_i under proactive mode;}else{    node_i under reactive mode;} if(receive HELLO){    node_i sends ID_i;    node_i sends hierarchy_i;}coordinator_node makes a REQUEST for each node;coordinator_node makes a hierarchy_table;coordinator_node chooses the best route from hierarchy_table;end

## 4. Results

The results presented in this section are divided into two parts. First, we present the results of the comparison and performance of the two systems proposed for the operation of sensor monitoring: cooperative and collaborative. Subsequently, we present the impact of the sensor system on the physiological perception of the people in our experiment. In this analysis, the collaborative scheme is used (because it is more versatile for the network) because it is transparent to the person how the system algorithm works. Finally, we add a usability analysis of the monitoring system.

### 4.1. Monitoring System Performance

The impact described in Figure 2 shows the difference in each specified metric between the network in a stable state without any technique (neither cooperation nor collaboration) and the network in a stable state with one of the two techniques applied. In this case, with the impact, we refer to each technique’s improvement. The metrics taken into account are related to the performance of the sensor network. So, for example, the size of the routing tables influences the number of entries a node has to send packets. It is related to the number of valid routes and the effort the routing algorithm has to make to find the next hop to a destination route. The overhead refers to the control packets, that is, to the packets that flood the network for management of the routing protocol. These packets, if there are too many, can generate more collisions than necessary because the network also has traffic packets, which are the ones that carry the information. The end-to-end delay is the delay that a packet takes from the source to the final destination. The Packet Delivery Ratio (PDR) is the rate of packets that reach the destination. It is the ratio between packets received and packets sent. Finally, the energy is given in Joules, which the node spends on the main network tasks such as turning on, turning off, route processing algorithm, changing state, transmitting and sending packets and listening to the channel.

Table 3 quantitatively analyzes some metrics related to the efficiency of the network concerning the user. Through this analysis, we could establish recommendations for cooperative or collaborative schemes for monitoring people according to the needs of the home or health center. So, for example, false positive alarms are higher in the collaborative scheme. This may be because, in the cooperative system, the nodes are more responsible for themselves, making them more successful in their tasks. Regarding resilience, the collaborative scheme is more efficient. This can be caused because resilience is a team effort where the network can return to its stable state quickly and when the nodes collaborate, they make this task faster. Regarding the delay in notifications or daily recommendations, the collaborative scheme is more efficient because the coordinator node has a more up-to-date perspective of the network topology, which means that the routes are always available. This metric matches the end-to-end delay in Figure 2 concerning the fastest scheme. Logical topology changes refer to how often the nodes change proactively or reactively and vice versa. This makes the network priorities set to a few nodes with priority (in proactive mode) because they notice more attention alerts. This metric is neither good nor bad, but it gives an idea of how the topology changes for one scheme or another. However, these changes can generate accurate alerts that a person has abruptly changed their habits and can be a genuine indication of an emergency.

Logical topology change refers to the way in which the communication pathways within a network are organized, as opposed to the physical layout of the devices. In wireless sensor networks for remote monitoring, logical topology change can have a significant impact on the cooperative and collaborative schemes used by the nodes. For example, if a node in the network fails or is taken offline, the logical topology of the network must be adjusted to account for the loss of that node. This can be a time-consuming and resource-intensive process, which can negatively impact the overall performance of the network.

Network resilience, on the other hand, refers to a network’s ability to maintain its functionality and performance even in the face of disruptions or failures. In wireless sensor networks for remote monitoring, network resilience is critical for ensuring that the nodes can continue to cooperate and collaborate effectively, even in the event of topology changes or other disruptions. By implementing robust network resilience strategies, such as redundant communication pathways and self-healing mechanisms, wireless sensor networks can minimize the impact of disruptions and maintain their performance and reliability.

Figure 3 shows the logical topology change over five days. This change is outlined for each scheme studied. We observe that, under the cooperative scheme, the network shows fewer links (30% approximately) concerning the collaborative scheme. In both cases, the system increases the number of logical links per node as the days go by. This is because the algorithm knows more routes, which have a lifetime in the routing table. Some of them are still valid and others have become obsolete. The greater the number of links (routes), the greater the redundancy and congestion. However, protocol management and the number of control packets can lead to more collisions. With fewer links in the nodes, the information delay and the network’s power consumption can decrease.

### 4.2. System Impact on the Person

We experimented for 2 weeks with the sensor monitoring system and 2 weeks without the system. We carried out tests in the homes of 50 people between 50 and 70 who live alone. Of these people, 23 were women and 27 were men.

Having some vital signs monitored can impact a person’s peace of mind by providing information about their overall health. Abnormal readings can indicate underlying health problems, such as high blood pressure, heart disease, respiratory problems or fever, which can cause stress and anxiety. On the other hand, having normal readings can give a person a sense of reassurance and peace of mind. Regular monitoring of these vital signs is important for maintaining good health and detecting potential problems early.

Heart rate (HR) or pulse: The heart rate, also known as pulse, is the number of times the heart beats per minute. It is usually measured by counting the number of beats in a minute at the wrist or neck.

Respiratory rate (RR): Respiratory rate is the number of breaths a person takes in a minute. It is usually measured by counting the number of breaths a person takes in one minute.

Temperature (T): Body temperature is a measure of the body’s ability to produce and get rid of heat. Normal body temperature is about 98.6 °F (37 °C).

Rapid Eye Movement Sleep (REMS): This is a stage of sleep characterized by quick movements of the eyes and increased brain activity. This stage is also known for vivid dreaming and decreased muscle tone, which prevents physical movement during sleep. REM sleep is an important part of the sleep cycle and typically occurs several times during a night’s sleep.

We report the REMS analysis through a smartwatch for a person’s commercial use and with this we analyze this metric in such a way that we can do it at home and check it every day upon waking up.

Based on the analysis of a person’s main vital signs, we analyzed their daily monitoring. This study considered healthy people (not high-performance athletes) who were non-smokers and had no cardiac or respiratory conditions.

For this experimentation, we consider the measurement daily and at the same time when the person wakes up. This is so that the person is not agitated or prone to worries during the day caused by external factors.

Figure 4 shows the behavior of most of the data with and without the monitoring system. We divided the population into men and women because there are physiological differences in heart rate according to gender. Men and women have differences in their anatomy, hormone levels and physical activity patterns, which contribute to variations in heart rate. Men generally have larger hearts and more muscle mass, resulting in a lower resting heart rate. Hormonal differences also play a role, with higher levels of testosterone in men leading to a lower heart rate than in women with higher levels of estrogen. Women also experience fluctuations in hormone levels during their menstrual cycle and pregnancy, which can affect their heart rate. Additionally, women tend to have less physically active lifestyles than men, leading to differences in heart rates between the two sexes. We obtained this information from the sensors because women mainly reported in specific areas of the house. This is accomplished by input from motion sensors and door gyroscopes. Conversely, men are reported to be more active in different areas of the house during the day, which could be associated with more activity.

We then found that, apparently, for women, the monitoring system represents a kind of greater peace of mind by 4% when they are under the presence of the monitoring system. Concerning men, this difference is only 2%. For both cases, in the presence of the sensor system, the heart rate numbers decrease for both genders.

Table 4 shows the change or variability a metric can have in the presence or absence of the sensor system. We find that the metric that varies the most is the Respiratory Rate at 13%. Subsequently, we have the Heart Rate at 5% and Temperature at 2%. This may lead us to think that the most conclusive metric of relationship with and without the sensor system is RR, with more strength in its variation for the majority of the sample.

After, we perform a correlation analysis between the metrics for both scenarios, with and without the use of the sensor network. At this point, to perform the appropriate correlation coefficient we examine if each separate metric follows a normal distribution or not. If a couple of metrics from the same scenario (with the sensor network or without the sensor network) follow a normal distribution we can use a parametric correlation coefficient. Otherwise, we have to use a non-parametric correlation coefficient. As in the paired hypothesis test analysis with the graphical methods it was difficult to determine if the metrics are normally distributed and we used a statistical normality test. Table 5 and Table 6 show the results of the Kolmogorov–Smirnov test for the nine metrics for each scenario.

The Spearman correlation coefficient is a non-parametric method that measures the strength and direction of the association between two variables. It is advantageous because it can be used with ordinal, interval or ratio data and does not require the variables to be normally distributed, making it robust to outliers and suitable for data with non-normal distributions. Additionally, the calculation of the Spearman coefficient can detect monotonic relationships, which are relationships that increase or decrease consistently but may not necessarily be linear. Therefore, the Spearman coefficient is a useful tool for examining correlations between pairs of metrics, particularly when the data are not normally distributed or when outliers are present.

These metrics do not follow the normal distribution with and without the system. We use the Spearman correlational coefficient to examine the correlations between each pair of metrics in both scenarios (with and without the sensor network). The following statistically significant correlations in the metrics without the use of the sensor network scenario were found:Temperature and heart rate present negative correlations of −0.331 with a *p*-value of 0.001.Temperature and REM sleep preset a positive correlation of 0.221 with a *p*-value of 0.027.Respiratory Rate and heart rate variability present a positive correlation of 0.211 with a *p*-value of 0.035.

With the sensor network the following statistically significant correlations were found:Temperature and REM sleep present a negative correlation of −0.314 with a *p*-value of 0.001.

The results of the correlation analysis revealed some trends, although the correlations are not strong. The negative correlation between total sleep time and snoring with the use of the sensor network suggests that elderly people slept more and with fewer interruptions. The correlation between temperature and REM sleep changed direction and became negative, indicating that REM sleep increases when temperature decreases. This study found a correlation between REM sleep and temperature, but not between deep sleep and temperature, as REM sleep occurs for a longer period during sleep. Finally, the use of the sensor network caused the correlations between heart rate and temperature and respiratory rate and heart rate variability to disappear. The correlation found above reflects abnormal behavior. When the sensor network is active, the correlation behavior is normal. So these metrics stabilize, statistically speaking, in the presence of the system.

### 4.3. Usability

In the design of any system, it is essential to reduce uncertainty, and relying on quantifiable data obtained in the investigation is beneficial. Usability is an important measure of understanding the user’s trust in the system. When we talk about health and, even more importantly, when health is monitored informally in our homes, it is necessary to have the user’s opinions to understand their needs. To consider user experience, we can measure whether the system helps or hinders the experience if the system improves over time and how the system compares to another that the user has known.

The usability of a sensor system installed in the home of an older adult is measured to ensure that the system is effective and easy to use for both the older adult and their caregivers or family members. Measuring usability can help identify any issues or barriers that may prevent the system from being used effectively and can help improve the overall quality and effectiveness of the system.

The benefits of measuring the usability of a sensor network that includes noise, motion, temperature, pressure, humidity, air quality, gyroscope and light sensors are numerous. Measuring usability can help ensure that the system is easy to use and understand and can help identify any issues that may prevent the system from working effectively. This can help improve the overall safety, security and wellbeing of older adults in their homes and can provide peace of mind for their caregivers and family members.

Measuring the usability of a non-invasive system for monitoring people at home also has significant benefits. Non-invasive systems are designed to be minimally intrusive and can help older adults maintain their independence and privacy while still receiving the care and support they need. Measuring usability can help ensure that these systems are effective and easy to use and can help identify any issues that may prevent the system from working effectively. This can help improve the overall quality of life for older adults and can provide peace of mind for their caregivers and family members [59].

We measured and asked users about their experience with the system installed in their homes. Efficacy answers whether users can achieve their goals with the system. To measure effectiveness, we defined what success and failure mean for the user concerning their relationship with the system. For example, we defined tasks such as turning on or off the sensor system and receiving advice by email at the beginning of the day. Efficiency in using the system refers to how much mental effort is required to perform a task. Here, users and their relatives receive alert messages and those under the network’s care receive recommendations as well. In most cases, the users’ satisfaction depends mainly on their ability to achieve their goals with minimum effort. The most direct way to assess satisfaction is through perceived usability and user experience questionnaires. Here, the satisfaction question is evaluated concerning the liking and security they feel with the sensor system and its performance.

For these results, we carried out a survey of five questions to users:Is it easy to activate and deactivate the sensor system?Does the system restore itself immediately when there is a temporary loss of electricity?Do the messages and recommendations arrive each day at the previously scheduled time?Is system reprogramming complicated and/or continuous?Are you comfortable with the sensor system installed in your home?

The effectiveness, efficiency and satisfaction metrics were measured on a Likert scale of percentages as in Figure 5. The first two questions are related to effectiveness, the next two are related to efficiency and the last is related to satisfaction. We carried out the same survey under the two proposed schemes. Although the user does not have to understand the algorithm’s intelligence, the network’s behavior reflects its consequences on the user and their interaction with the system. We found that the collaborative system is more pleasant and shows 8% better performance for the user.

## 5. Discussion

Remote monitoring systems with sensor networks in smart homes for the elderly have the potential to provide a non-invasive way to detect and respond to potential emergencies. One of the key elements of these systems is the use of routing protocols in the sensor devices. These protocols are used to ensure that the data collected by the sensors are transmitted to the central hub or cloud-based platform in a timely and efficient manner. This is critical for ensuring that the data are available for analysis and can be used to detect potential issues or changes in the individual’s health or wellbeing.

One of the most important aspects of these routing protocols is their ability to adapt to changing conditions in the network. For example, if a sensor goes offline or the network becomes congested, the routing protocol will automatically adjust to ensure that the data continues to be transmitted. This can be especially important in smart homes for the elderly, where the individual may be at risk of falls or other accidents and prompt intervention is critical. The use of routing protocols that are designed to be robust and reliable can help to ensure that the data is transmitted even in the event of a network failure.

Another important aspect of routing protocols in remote monitoring systems is their ability to optimize the use of network resources. This can be achieved by using algorithms that are designed to minimize the energy consumption of the sensor devices, reduce the amount of data transmitted over the network or improve the overall efficiency of the system. These algorithms can be especially important in smart homes for the elderly, where the individual may have limited mobility and may be unable to replace batteries or other power sources. By optimizing the use of network resources, these algorithms can help to prolong the battery life of the sensor devices and reduce the need for maintenance.

Overall, remote monitoring systems with sensor networks in smart homes for the elderly have the potential to provide a non-invasive, efficient and reliable way to detect and respond to potential emergencies. The use of routing protocols that are designed to be robust, adaptive and energy-efficient can help to ensure that the data are transmitted in a timely and efficient manner and that the system can adapt to changing conditions in the network. Additionally, these protocols are able to optimize the use of network resources, prolong the battery life of the sensor devices and reduce the need for maintenance, which could be critical for elderly individuals who may have limited mobility.

## 6. Conclusions

Monitoring systems with sensor networks in smart homes for the elderly have the potential to provide a non-invasive way to detect and respond to potential emergencies while also providing a user-friendly and efficient experience. These systems can be used in a variety of applications, such as monitoring vital signs, tracking movements and activities and detecting potential fall risks.

One of the key applications of these systems is in the monitoring of vital signs, such as heart rate and oxygen saturation. This can provide early warning signs of potential health issues, allowing for prompt intervention and medical attention. This can be especially important for older adults who may be at risk of falls or other accidents, as well as those with chronic conditions such as heart disease or diabetes.

Another important application of these systems is in tracking movements and activities. By monitoring the individual’s activities and movements, the systems can identify patterns of behavior that can be used to adjust the individual’s care plan. This can include adjusting the lighting or temperature in the home or providing reminders to take medication or perform other important tasks.

Furthermore, these systems can also be used to detect potential fall risks. By monitoring the individual’s movements and vital signs, the system can detect changes that may indicate a decline in mobility or a fall risk. This can include changes in gait, changes in heart rate or oxygen saturation or changes in sleep patterns.

Usability is also an important aspect of these systems, as it can greatly affect the adoption and effectiveness of the system. A system that is easy to use, provides clear information and offers an overall positive experience can increase the adoption and effectiveness of the system. Additionally, a user-friendly system can ensure that the data are collected and transmitted in a timely and accurate manner and that the system is able to detect potential issues or changes in the individual’s health or wellbeing.

Overall, monitoring systems with sensor networks in smart homes for the elderly have the potential to provide a non-invasive, efficient and user-friendly way to detect and respond to potential emergencies. These systems can be used in a variety of applications, such as monitoring vital signs, tracking movements and activities and detecting potential fall risks. By designing systems with a user-centered approach, it is possible to create systems that are easy to use and that provide clear information, which can increase the adoption and effectiveness of the system.

The authors’ work on wireless sensor networks in healthcare is an important contribution to the field. Their study successfully achieved the goal of comparing two working schemes in the routing protocol algorithm of WSNs for home monitoring of elderly patients. The authors demonstrated that the cooperative scheme is optimal in terms of speed of information delivery, network resilience, stability of communications and energy consumption. Furthermore, their statistical analysis of the impact of the sensor network on vital signs provides valuable insights for healthcare professionals. This research provides a foundation for developing more efficient and reliable WSNs for elderly care, improving the quality of life of elderly people who wish to maintain their independence at home. The authors’ work is an original and novel way of optimizing the performance of the algorithm and the ease of use for people, with a focus on improving the quality of care for patients.

## Figures and Tables

**Figure 1 ijerph-20-05268-f001:**
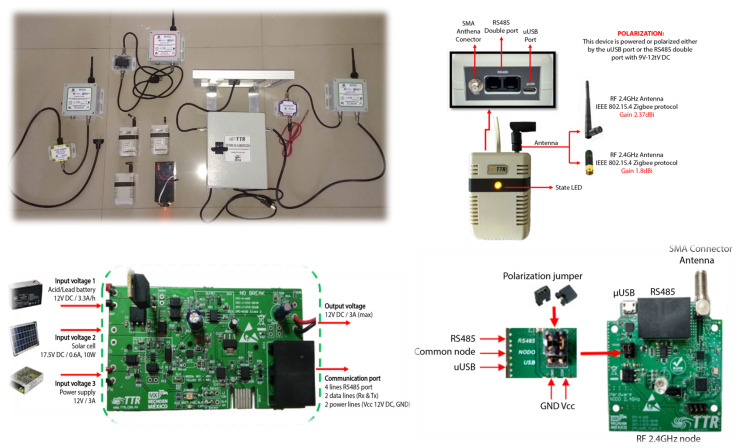
Devices of the network system.

**Figure 2 ijerph-20-05268-f002:**
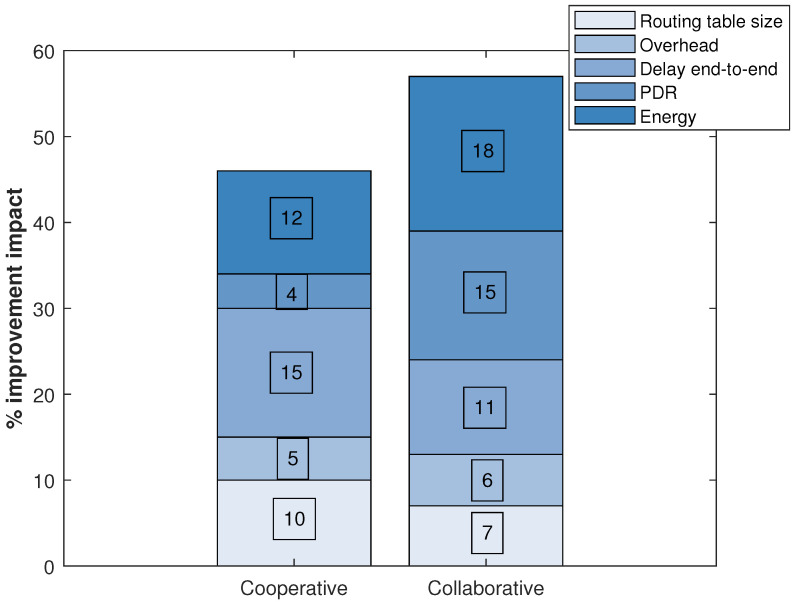
Percentage of improvement impact on network performance metrics.

**Figure 3 ijerph-20-05268-f003:**
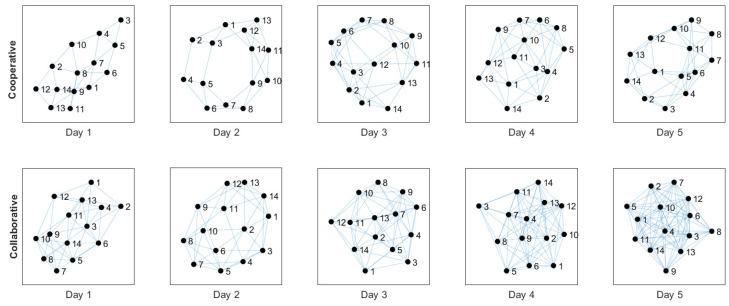
Logical change of topology through 5 days under each system scheme.

**Figure 4 ijerph-20-05268-f004:**
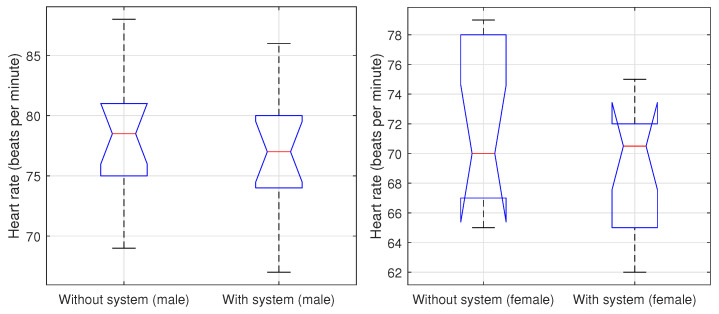
Heart rate with and without the monitoring system for men and women.

**Figure 5 ijerph-20-05268-f005:**
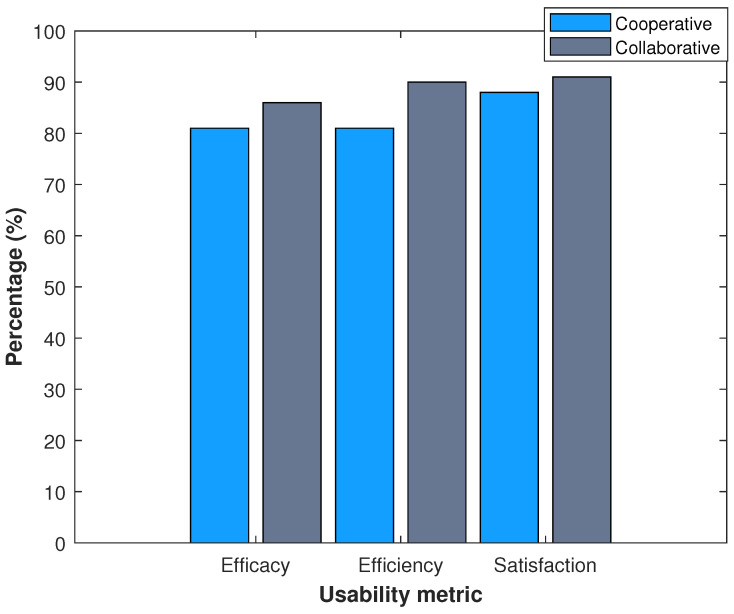
Measurement of usability metrics.

**Table 1 ijerph-20-05268-t001:** Comparison of related works and current paper references on monitoring of older people with wireless sensor networks.

Reference	Monitoring System	Wireless Technology Used	Measured Parameters	Experiment Time	Vital Sign Analysis
[36]	Wearable	Bluetooth Low Energy	Heart rate, respiration rate	1 day	No
[37]	In-home	ZigBee	Activity, fall detection	1 year	No
[38]	In-home	ZigBee	Activity, sleep, medication adherence	3 months	No
[39]	Wearable	Wi-Fi	Gait, fall risk	1 week	No
[40]	In-home	Wi-Fi	Activity, sleep	6 months	No
[41]	In-home	ZigBee	Blood pressure, heart rate, activity	6 months	Yes
[42]	In-home	Bluetooth Low Energy	Blood pressure, heart rate, weight	1 year	Yes
[43]	In-home	ZigBee	Activity, heart rate, respiration rate	6 months	Yes
[44]	In-home	Bluetooth Low Energy	Activity, sleep, gait	3 months	Yes

**Table 2 ijerph-20-05268-t002:** System network technical specifications.

Sensor	Features
Motion	DC 4.5–20 V, 50 μA delay: 5–200 adjustable, Operation Temp.: −15–+70 degrees, Detection Range: 3 m to 7 m.
Pressure, temperature and humidity	Combines thermometer, barometer and hygrometer. Temperature range from −40 to +85 ${}^{\circ }$C with an accuracy of ±1 ${}^{\circ }$C and resolution of 0.01 ${}^{\circ }$C and for pressure 300–1100 hPa, accuracy of ±1 Pa and resolution of 0.18 Pa. Supply voltage range: 1.71 V to 3.6 V. Accuracy tolerance ±3% relative humidity. Current Consumption 0.4 mA.
Noise	ULTRASONIC SENSOR HRXL-MAXSONAR. MAX4466 with adjustable gain with a resolution of 1 mm to 1 cm. 20–20 KHz electric microphone. 2.4–5 VDC. 3.7 W. Frequency: 42 kHz. Type: Transmitter, Receiver. Maximum detection distance: 765 cm. Consumption: 2.1 mA.
Light	LDR photoresistor sensor module LM393. Voltage: 3.3–5 V. Output Type: Digital. Propagation delay time (µs) 1.3. Faster response time of 1 µsec
Gyroscope	7A994. Axis Type: Single. Typical Angular Velocity (°/s): ±300. Typical Operating Supply Voltage (V): 3.3|5. Minimum Operating Temperature (${}^{\circ }$C): −40. Maximum Operating Temperature (${}^{\circ }$C): 105. Linearity: No.
Air quality	ZPHS01C Multi-in-One Air quality monitoring Sensor Module. Target Gas: PM2.5, CO2, CH2O, TVOC, Temperature and Humidity. Applications: Gas detector, Air conditioner, Air quality monitoring, Air purifier, HVAC system, Smart home. Size: 62.5 mm (L) × 61 mm (W) × 25 mm (H). Output signal UART/RS485.

**Table 3 ijerph-20-05268-t003:** System performance metrics towards the user.

Metric	Cooperative	Collaborative
False positive alarms	3%	5%
System resilience to loss of energizing	8 s	6 s
Accurate delivery of daily recommendations (delay)	2 s	1 s
System logical topology change per day	3 times	2 times

**Table 4 ijerph-20-05268-t004:** Relationship of metrics with and without the sensor system.

Metric	Metric Average without System	Metric Average with System
HR	73.51	69.88
RR	16.90	14.59
T	37.33	36.84

**Table 5 ijerph-20-05268-t005:** Results of Kolmogorov–Smirnov test for metrics without the system.

Metric	*p*-Value	Normal Distribution
HR	0.021	No
RR	0.037	No
T	0.000	No
REMS	0.010	No

**Table 6 ijerph-20-05268-t006:** Results of Kolmogorov–Smirnov test for metrics with the system.

Metric	*p*-Value	Normal Distribution
HR	0.002	No
RR	0.000	No
T	0.000	No
REMS	0.010	No

## Data Availability

The data presented in this study are available in Appendix A.

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
