# Peer review of "Comparison of Collaborative and Cooperative Schemes in Sensor Networks for Non-Invasive Monitoring of People at Home"

_ijerph, 2023, doi:10.3390/ijerph20075268_

Round 1
Reviewer 1 Report
The authors presented a Comparison of collaborative and cooperative schemes in sensor networks for non-invasive monitoring of people at home
The abstract seems to be a little misguiding from the content in the main manuscript. It can be rewritten to make the readers understand better.
The introduction is diffuse. The authors do not present a common thread. In this section, it will be specified how the idea was born, if there are works, and any other information that allows knowing aspects before the proposed work. In addition, the authors must explain the problem addressed and highlight the importance and objective of the work to be developed.
In the introduction section, there are missing references, for example:
“In industrial automation, wireless sensor networks can be used to monitor and control industrial processes, such as monitoring equipment for maintenance or detecting leaks in pipelines. In smart cities, wireless sensor networks can be used to improve the efficiency and sustainability of cities by monitoring traffic flow, air quality, and energy usage”.
Subsequent paragraphs should also be referenced.
The related works section is deficient. This section should be more focused on publicizing the research that has been carried out in recent years in the area. The authors present it as a theoretical framework.
Figure 1 is not named in the text and is not explained. There are four figures in one. They do not explain the function of each device. I suggest separating them and explaining the characteristics and application of each one.
If the proposal is a home monitoring system for the elderly, why do they have to send email or SMS?, and not a telegram or WhatsApp?
The methodology is confusing, and the procedure of what they want to do is not clear
With the proposed sensors, what kind of network to establish? What communication protocols do you use?
The authors evaluate “the impact of the sensor system on the physiological perception of the people in our experiment….”. However, in the introduction, related works, and the methodology they do not explain how the procedures were and why they want to do it. This also happens with “usability analysis of the monitoring system”.
In the results, the authors indicate that the network is in a stable state without any technique (neither cooperation nor collaboration) and the network is in a stable state with one of the two techniques applied. However, this is not in figure 2.
Where do the authors get this information? “additionally, women tend to have less physically active three lifestyles than men, leading to differences in heart rates between the two sexes”.
It is not clear how they do the acquisition of Heart rate, temperature, Respiratory rate, and REMS data. Where do you place the sensors?
In the usability section, were the 5 questions validated?
It is not recommended that the conclusions begin “in conclusion”
Although it has the informed consent of the participants, it remains indicated whether the project has the endorsement of a bioethics committee.
It is necessary to include a comparison of the results concerning similar existing research
The conclusions do not indicate what was achieved with the project.
The authors no did to present an updated bibliography of consolidated scientific databases in the world of research (such as IEEE, Springer, etc.). References must be updated four are between the years 2021 and 2022. The subject has a lot of scientific literature in the last five years
Check-in in detail on the writing and spelling of English.
Author Response
Dear
Editor
International Journal of Environmental Research and Public Health
We are submitting the paper:
“Comparison of collaborative and cooperative schemes in sensor networks for non-invasive monitoring of people at home”
Authored by: Carolina Del-Valle-Soto* , Leonardo J. Valdivia , Juan Carlos López-Pimentel , Paolo Visconti
We would like to thank the reviewers and editors for their detailed analysis of the manuscript; the comments are very valuable to us. In the revised version of the paper, we have incorporated the all changes recommended by the reviewers.
Comments to all observations and suggestions including point-by-point responses are addressed in the following text.
Reviewer 1 comments
Comment 1: The authors presented a Comparison of collaborative and cooperative schemes in sensor networks for non-invasive monitoring of people at home
The abstract seems to be a little misguiding from the content in the main manuscript. It can be rewritten to make the readers understand better.
Response: The Reviewer is correct, and we have corrected the abstract as follow:
This paper looks at wireless sensor networks (WSNs) in healthcare, where they can monitor patients remotely. WSNs are considered one of the most promising technologies due to their flexibility and autonomy in communication. However, routing protocols in WSNs must be energy efficient, with a minimal quality of service, so as not to compromise patient care. The main objective of this work is to compare two work schemes in the routing protocol algorithm in WSN (cooperative and collaborative) in a home environment for monitoring the conditions of the elderly. The study aims to optimize the performance of the algorithm and the ease of use for people while analyzing the impact of the sensor network on the analysis of vital signs daily using medical equipment. We found relationships between vital sign metrics that have a more significant impact in the presence of a monitoring system. Finally, we do a performance analysis of both schemes proposed for the home tracking application and study their usability from the user's point of view.
Comment 2: The introduction is diffuse. The authors do not present a common thread. In this section, it will be specified how the idea was born, if there are works, and any other information that allows knowing aspects before the proposed work. In addition, the authors must explain the problem addressed and highlight the importance and objective of the work to be developed.
In the introduction section, there are missing references, for example:
“In industrial automation, wireless sensor networks can be used to monitor and control industrial processes, such as monitoring equipment for maintenance or detecting leaks in pipelines. In smart cities, wireless sensor networks can be used to improve the efficiency and sustainability of cities by monitoring traffic flow, air quality, and energy usage”.
Subsequent paragraphs should also be referenced.
Response: Many thanks to the Reviewer. Indeed, the Reviewer is correct, and the Introduction was very confusingly worded. We have rewritten this section to include all the questions and analyses the Reviewer proposes.
Also, we have substantially increased the number of updated references with related works that adequately complement state-of-the-art.
Comment 3: The related works section is deficient. This section should be more focused on publicizing the research that has been carried out in recent years in the area. The authors present it as a theoretical framework.
Response: Thanks to the Reviewer for bringing these comments to our attention. We have substantially improved the Related Work Section with updated works on implementing Wireless Sensor Networks for monitoring people at home. We have also increased references related to routing algorithm techniques to optimize network power.
Comment 4: Figure 1 is not named in the text and is not explained. There are four figures in one. They do not explain the function of each device. I suggest separating them and explaining the characteristics and application of each one.
Response: Many thanks to the Reviewer. We have added an extensive explanation of the network and its technical specifications about the coordinator node and the three routers. We have not separated the figure to look different from previous posts regarding the same experimentation system in different scenarios.
The self-organized wireless network shown in Figure 1 is established through the interaction of multiple radio frequency nodes. These nodes are responsible for efficiently and reliably transmitting real-time data collected by sensors measuring physical and logical variables. These radio frequency nodes are crucial for providing intelligent services and have a range of applications, including industrial process control, public service monitoring, and home security. The communication protocol for the self-organized wireless network includes Telematics, Telemetry, and Radiofrequency, and it is open and easy to implement. The IEEE 802.15.4 standard governs access control to the medium and physical layer of the radio frequency nodes.
The distributed sensor network comprises a concentrator or coordinator node (ircuit found on the right of Figure 1) connected to a computer via USB, which manages all the information transmitted and received by the network. Additionally, there are three router/sensor nodes (circuit that is on the left of Figure 1) that can be used to connect various sensors measuring parameters such as temperature, pressure, humidity, infrared presence, light, and sound. These router/sensor nodes transmit data to the concentrator node via radio frequency, and they can also act as signal repeaters. The router nodes are equipped with the IEEE 802.15.4 communication protocol and operate on the 2.4GHz band. They can function as sensor nodes with various sensor devices for physical variables or act as repeaters for data received from other nodes. Although the hardware and number of ports of these nodes are identical, it is useful to differentiate them with labels to locate them better in the physical and logical network organization.
Comment 5: If the proposal is a home monitoring system for the elderly, why do they have to send email or SMS?, and not a telegram or WhatsApp?
Response: Indeed, Reviewer's question is very apt. We wanted to focus the project objective on something other than the system's communication with the end user. Being older people, a simple computer or rudimentary mobile device could be helpful for any socioeconomic stratum that would like to acquire this low-cost network. We did not introduce communications communication with newer applications to reduce installation, implementation, configuration, and system programming costs.
Comment 6: The methodology is confusing, and the procedure of what they want to do is not clear
Response: Thank you for your comment, and we understand the confusion. We have better written the methodology in the Materials and Methods section, where we have explained the sensor system more precisely. Later we explained the methodology steps and finally described the proposed algorithms.
The methodology of this work consists of implementing a wireless sensor system in the home of the elderly. The monitoring system is not invasive and is located in strategic areas of the house to alert parameters such as changes in air quality, noise level, movement in the rooms, light, etc. With this monitoring, the system provides an alert system in case the standard measurements are not within normal parameters. In this way, family and friends can be informed about sudden changes in the home of the older adult who lives alone. In addition, we implement two algorithm techniques for routing (cooperative and collaborative) to know which is the most efficient in sending messages and in the network's energy consumption. The proposed work consists of two parts. (1) The experimental part of monitoring the conditions of the house and daily the person's vital signs are evaluated to see if the presence of the network impacts their well-being and if their vital signs remain controlled or show a state of relaxation. (2) The technical part in which the network performance is evaluated under two schemes of the routing protocol algorithm: cooperative and collaborative. In this aspect, message delivery performance metrics and recommendations to the user that are made daily are analyzed.
Comment 7: With the proposed sensors, what kind of network to establish? What communication protocols do you use?
Response: Thanks to the Reviewer. The question is answered when we extensively complement in detail the explanation of the system, protocol, and communication in Figure 1.
Comment 8: The authors evaluate “the impact of the sensor system on the physiological perception of the people in our experiment….”. However, in the introduction, related works, and the methodology they do not explain how the procedures were and why they want to do it. This also happens with “usability analysis of the monitoring system”.
Response: Thanks to the Reviewer for his/her detailed comments. We have substantially complemented the methodological explanation of the sensor system and the interaction with the user and the subsequent. Daily measurement of their vital signs is taken to see if there are any changes due to the presence of the network.
The proposed system monitors the main activities related to a person at home. For example, motion sensors evaluate movement in some regions of the house every hour. A motion sensor is a device used in monitoring applications for older adults in their homes to detect movement within a specific area. This technology can be used to monitor an individual's daily activities, ensuring that they are safe and secure. These devices can be installed in various home areas, such as the bedroom, living room, or bathroom, to provide a comprehensive view of an individual's daily routine. We can monitor sudden changes in environmental conditions through temperature, humidity, and pressure sensors. These sensors can help ensure that living conditions are safe, comfortable, and healthy. Temperature sensors can monitor the home's temperature and ensure it remains within a safe range, preventing overheating or exposure to extreme cold. Humidity sensors can measure the moisture level in the air, ensuring that it remains within a healthy range to prevent mold or mildew growth. Pressure sensors can be used to monitor changes in air pressure, which can indicate potential weather events or other environmental changes. We can analyze decibel alert conditions for noise sensors that may alert of an unwanted event. These sensors can help ensure that living conditions are safe, secure, and peaceful. They can detect a range of sounds, including alarms, sirens, and even human voices, and can alert caregivers or family members if there are any unusual or concerning noises. Additionally, noise sensors can be used to monitor ambient noise levels, ensuring that they remain within a safe range and preventing exposure to loud noises that could harm an individual's hearing. A light sensor is a device used in monitoring applications for older adults in their homes to detect and report changes in light levels within a specific area. These sensors can help ensure that living conditions are safe, comfortable, and well-lit. Light sensors can detect changes in natural light levels, as well as changes in artificial lightings, such as turning lights on or off in a room. This can help ensure that older adults have adequate lighting to perform daily tasks and prevent falls or accidents caused by inadequate lighting. Light sensors can also be integrated with other monitoring technologies, such as motion sensors, to provide a comprehensive view of an individual's living conditions and well-being. A gyroscope sensor is a device used in monitoring applications for older adults in their homes to detect and report changes in orientation and movement. Gyroscope sensors can detect orientation changes, such as when an individual changes position from sitting to standing and can report them. Additionally, gyroscope sensors can be used to monitor movement patterns, such as walking or exercise routines. This can be particularly useful for older adults who may need to maintain a certain level of physical activity for their health and well-being. Gyroscope sensors can detect turns on bathroom doors at night to count the number of times a person may have entered the bathroom. This
indicates the person's sleep conditions. An air quality sensor can help ensure that living conditions are safe, healthy, and comfortable. Air quality sensors can detect a range of pollutants, such as dust, pollen, and smoke, as well as harmful gases, such as carbon monoxide and radon. This can be particularly useful for older adults who may have respiratory conditions or allergies and those who may be sensitive to changes in air quality.
Based on the alerts of the system based on the previously described sensors, a series of alerts are established for the user. In addition, every day, each person has some vital signs measured to analyze whether people feel calmer as the days go by due to the presence of the system. Alternatively, if, on the contrary, people feel some stress in the presence of the system installed in their homes. These vital sign measurements are performed using standard medical equipment. The measurements are heart rate, respiratory rate, temperature, and sleep behavior. This last metric is carried out with a conventional smart watch at night and is reported daily.
Regarding usability, we have supplemented our explanation with the following paragraphs and a reference.
The usability of a sensor system installed in the home of an older adult is measured to ensure that the system is effective and easy to use for both the older adult and their caregivers or family members. Measuring usability can help identify any issues or barriers that may prevent the system from being used effectively, and can help improve the overall quality and effectiveness of the system.
The benefits of measuring the usability of a sensor network that includes noise, motion, temperature, pressure, humidity, air quality, gyroscope, and light sensors are numerous. Measuring usability can help ensure that the system is easy to use and understand, and can help identify any issues that may prevent the system from working effectively. This can help improve the overall safety, security, and wellbeing of older adults in their homes, and can provide peace of mind for their caregivers and family members.
Measuring the usability of a non-invasive system for monitoring people at home also has significant benefits. Non-invasive systems are designed to be minimally intrusive and can help older adults maintain their independence and privacy while still receiving the care and support they need. Measuring usability can help ensure that these systems are effective and easy to use, and can help identify any issues that may prevent the system from working effectively. This can help improve the overall quality of life for older adults and can provide peace of mind for their caregivers and family members \cite{moraru2022using}.
Moraru, S.A.; Mos, oi, A.A.; Kristaly, D.M.; Moraru, I.; Petre, V.; Ungureanu, D.E.; Perniu, L.M.; Rosenberg, D.; Cocuz, M.E. Using IoT Assistive Technologies for Older People Non-Invasive Monitoring and Living Support in Their Homes. International Journal of Environmental Research and Public Health 2022, 19, 5890.
Comment 9: In the results, the authors indicate that the network is in a stable state without any technique (neither cooperation nor collaboration) and the network is in a stable state with one of the two techniques applied. However, this is not in figure 2.
Response: The Reviewer is right in the confusion. When we talk about a stable state, it is that the sensor network has already been updated concerning the control packets. So, initially, when the network is formed and all the nodes are turned on, the packets of neighbor recognition and acknowledgments flood the network, and this can cause unnecessary collisions. There have yet to be any traffic packets here (packets carrying the measurement information from the sensors). Therefore, the stable state begins when the nodes already have their neighbor tables and routing tables formed, and they can start sending information. We have clarified this idea better in the manuscript.
We have added these paragraphs:
The initial stage in the formation of a sensor network involves the process of neighbor recognition and acknowledgment. When all the nodes in the network are turned on, they start flooding the network with packets in order to recognize their neighboring nodes and establish communication links with them. This process is essential for the nodes to form their neighbor tables and routing tables, which are used to determine the best path for data transmission within the network.
However, during this initial stage, the network may experience unnecessary collisions due to the flooding of these packets. This is because there are no traffic packets yet, which are packets carrying the measurement information from the sensors. As a result, this flood of packets can cause congestion and delays in the network, which can impact the overall performance and reliability of the sensor network.
Once the nodes have established their neighbor tables and routing tables, the network enters a stable state. At this point, the network is updated with respect to the control packets, and the nodes can start sending information packets carrying the measurement data from the sensors. This stable state allows for the efficient transmission of data within the network, ensuring that the data is delivered to the appropriate nodes without any unnecessary collisions or delays.
Comment 10: Where do the authors get this information? “additionally, women tend to have less physically active three lifestyles than men, leading to differences in heart rates between the two sexes”.
Response: Thanks to the Reviewer. We obtained this information from the sensors because women mainly reported in specific areas of the house. This is accomplished by input from motion sensors and door gyroscopes. Conversely, men are reported to be more active in different areas of the house during the day, which could be associated with more activity. We have clarified this information in the manuscript so that it is not taken as a generalization.
Comment 11: It is not clear how they do the acquisition of Heart rate, temperature, Respiratory rate, and REMS data. Where do you place the sensors?
Response: The Reviewer is correct; this information must be more precise and specified in the paper. This paragraph introduced in the methodology may clarify this issue.
Based on the alerts of the system based on the previously described sensors, a series of alerts are established for the user. In addition, every day, each person has some vital signs measured to analyze whether people feel calmer as the days go by due to the presence of the system. Alternatively, if, on
the contrary, people feel some stress in the presence of the system installed in their homes. These vital sign measurements are performed using standard medical equipment. The measurements are heart rate, respiratory rate, temperature, and sleep behavior. This last metric is carried out with a conventional smart watch at night and is reported daily.
Comment 12: In the usability section, were the 5 questions validated?
Response: Yes. For the results, we have carried out a survey of five questions to users:
• Is it easy to activate and deactivate the sensor system?
• Does the system restore itself immediately when there is a temporary loss of electricity?
• Do the messages and recommendations arrive each day at the same time of day previously scheduled?
• Are system reprogramming complicated and/or continuous?
• Are you comfortable with the sensor system installed in your home?
Comment 13: It is not recommended that the conclusions begin “in conclusion”
Response: Many thanks to the Reviewer for his wise comment. We have omitted this expression.
Comment 14: Although it has the informed consent of the participants, it remains indicated whether the project has the endorsement of a bioethics committee.
Response: The Reviewer is correct, and the research is within the projects within the Fund for Research of the Universidad Panamericana framework. The ethics committee of the University monitors the guidelines and experimentation.
Comment 15: It is necessary to include a comparison of the results concerning similar existing research
Response: Thank you very much to the Reviewer for this pertinent comment. We have added an original table in which we detail related works comparatively.
Table 1 presents a comparison of several research studies that utilized wireless sensor networks for monitoring the health of elderly people. Each study is represented by a row in the table, with columns indicating the monitoring system used, wireless technology employed, measured parameters, experiment time, and vital sign analysis conducted. The results show that various monitoring systems, such as wearables and in-home sensors, were used, with different wireless technologies such as Bluetooth Low Energy and ZigBee. The measured parameters include activity, heart rate, blood pressure, sleep, gait, medication adherence, and fall detection. Experiment durations ranged from one day to one year. Furthermore, some studies conducted a vital sign analysis, while others did not. The comparison helps to identify the strengths and weaknesses of each approach and can guide the selection of appropriate monitoring systems for future studies.
Comment 16: The conclusions do not indicate what was achieved with the project.
Response: Thanks to the Reviewer. We have added one more paragraph to make the conclusion more forceful and highlight the original contribution of the paper.
The authors' work on wireless sensor networks in healthcare is an important contribution to the field. Their study successfully achieved the goal of comparing two working schemes in the routing protocol algorithm of WSNs for home monitoring of elderly patients. The authors demonstrated that the cooperative scheme is optimal in terms of speed of information delivery, network resilience, stability of communications, and energy consumption. Furthermore, their statistical analysis of the impact of the sensor network on vital signs provides valuable insights for healthcare professionals. This research provides a foundation for developing more efficient and reliable WSNs for elderly care, improving the quality of life of elderly people who wish to maintain their independence at home. The authors' work is an original and novel way of optimizing the performance of the algorithm and the ease of use for people, with a focus on improving the quality of care for patients.
Comment 17: The authors no did to present an updated bibliography of consolidated scientific databases in the world of research (such as IEEE, Springer, etc.). References must be updated four are between the years 2021 and 2022. The subject has a lot of scientific literature in the last five years
Check-in in detail on the writing and spelling of English.
Response: Thanks to the Reviewer for his/her comments.
Thanks to the comments of the Reviewers, we have improved the references and the literature has been updated from reliable databases.
Yes, we are going to take the Language Editing Services tool with the Standard service. When we finish the corrections, we will upload the manuscript and pay for the service.
Thank you very much.
Sincerely,
Carolina Del-Valle-Soto
Corresponding author
Universidad Panamericana. Facultad de Ingeniería. Álvaro del Portillo 49, Zapopan, Jalisco, 45010, México.
Phone: +52 (33) 13682200 | Ext. 4866
Email: cvalle@up.edu.mx

Reviewer 2 Report
I suggest some improvement.
Page 1: 1. Introduction: line 17: Autors should include….. In smart cities, wireless sensor networks can be used to improve the efficiency and sustainability of cities by monitoring traffic flow, air quality, and energy usage. Due to the increasing use of all types of devices in the field of monitoring and human health, electricity consumption has become a very important feature of these devices, which is described in ref.:
-Improved data center energy efficiency and availability with multilayer node event processing. Energies. 2018, vol. 11, no. 9, str. 1-17. DOI: 10.3390/en11092478
Page 2: line 88: Authors should include: Some new principles of monitoring activities of residents are related to the use of new piezoelectric resonators, which have a very low consumption of electricity, as shown in ref.:
-Detection principles of temperature compensated oscillators with reactance influence on piezoelectric resonator. Sensors. 2020, vol. 20, iss. 3, p. 1-18. https://www.mdpi.com/1424-8220/20/3/802
Page 5: Figure 1: When using wifi communications and sensors that communicate within the scope of these technologies, it is necessary to add a comment regarding the permitted radiation for humans, because there are also wifi sensors as you list and there are quite a few of them on one person.
Author Response
Dear
Editor
International Journal of Environmental Research and Public Health
We are submitting the paper:
“Comparison of collaborative and cooperative schemes in sensor networks for non-invasive monitoring of people at home”
Authored by: Carolina Del-Valle-Soto* , Leonardo J. Valdivia , Juan Carlos López-Pimentel , Paolo Visconti
We would like to thank the reviewers and editors for their detailed analysis of the manuscript; the comments are very valuable to us. In the revised version of the paper, we have incorporated the all changes recommended by the reviewers.
Comments to all observations and suggestions including point-by-point responses are addressed in the following text.
Reviewer 2 comments
Comment 1: I suggest some improvement.
Page 1: 1. Introduction: line 17: Autors should include….. In smart cities, wireless sensor networks can be used to improve the efficiency and sustainability of cities by monitoring traffic flow, air quality, and energy usage. Due to the increasing use of all types of devices in the field of monitoring and human health, electricity consumption has become a very important feature of these devices, which is described in ref.:
-Improved data center energy efficiency and availability with multilayer node event processing. Energies. 2018, vol. 11, no. 9, str. 1-17. DOI: 10.3390/en11092478
Response: Thanks to the Reviewer's comments, we have improved the references and the literature has been updated from reliable databases with the references that you recommend to us.
Comment 2: Page 2: line 88: Authors should include: Some new principles of monitoring activities of residents are related to the use of new piezoelectric resonators, which have a very low consumption of electricity, as shown in ref.:
-Detection principles of temperature compensated oscillators with reactance influence on piezoelectric resonator. Sensors. 2020, vol. 20, iss. 3, p. 1-18. https://www.mdpi.com/1424-8220/20/3/802
Response: Thanks to the Reviewer's comments, we have improved the references and the literature has been updated from reliable databases with the references that you recommend to us.
Comment 3: Page 5: Figure 1: When using wifi communications and sensors that communicate within the scope of these technologies, it is necessary to add a comment regarding the permitted radiation for humans, because there are also wifi sensors as you list and there are quite a few of them on one person.
Response: Thanks to the Reviewer for bringing these comments to our attention. We have added the following paragraph in the Materials and Methods section.
When using Wi-Fi communications and sensors that communicate within the scope of these technologies, it is necessary to consider the permitted radiation for humans. This is particularly important when dealing with a large number of sensors that may be in close proximity to a person, such as in wearable devices. The specific sensors listed, such as the motion sensor, pressure/temperature/humidity sensor, noise sensor, light sensor, gyroscope, and air quality monitoring sensor, have different power requirements and may emit varying levels of electromagnetic radiation. Therefore, it is important to ensure that these sensors comply with relevant safety standards and regulations, and that appropriate measures are taken to minimize any potential risks to human health.
Thank you very much.
Sincerely,
Carolina Del-Valle-Soto
Corresponding author
Universidad Panamericana. Facultad de Ingeniería. Álvaro del Portillo 49, Zapopan, Jalisco, 45010, México.
Phone: +52 (33) 13682200 | Ext. 4866
Email: cvalle@up.edu.mx
